# Blue Nevi and Melanoma Arising in Blue Nevus: A Comparative Histopathological Case Series

**DOI:** 10.3390/reports8030131

**Published:** 2025-08-01

**Authors:** Hristo Popov, Pavel Pavlov, George S. Stoyanov

**Affiliations:** 1Department of General and Clinical Pathology, Forensic Medicine and Deontology, Faculty of Medicine, Medical University—Varna, 9002 Varna, Bulgaria; 2Department of Pathology, Complex Oncology Center, 9700 Shumen, Bulgaria; 3Department of Pathology, Multiprofile Hospital for Active Treatment, 9700 Shumen, Bulgaria

**Keywords:** nevus, melanocytic lesion, blue nevus, cellular blue nevus, sclerotic dendritic blue nevus, dendritic blue nevus, melanoma arising in blue nevus

## Abstract

**Background and Clinical Significance:** Blue nevi are a dubious pigmented lesion. While somewhat common throughout the population, they are significantly less common than other melanocytic neoplasms, and both their morphology and development bring them closer to true hamartomas than neoplasms. An exceedingly rare occurrence is the development of melanoma from a preexisting blue nevus. This nosological unit, defined as melanoma arising in a blue nevus, also known as malignant blue nevus, blue naevus–like melanoma, melanoma ex-blue naevus, and melanoma mimicking cellular blue naevus, is required to either originate from an area of previously excised blue nevus or have a blue nevus remnant adjacent to it. Due to the spindle cell morphology of melanoma arising in blue nevus, the terminology is often misused by some authors to include spindle cell melanomas, which exhibit a distinct pathogenesis and, although morphologically similar, have differing molecular profiles as well. **Case presentations:** The following manuscript discusses comparative morphological features in a case series of blue nevi and melanoma arising in blue nevi. **Discussion:** Blue nevi present with unique morphological features, with melanomas originating from them having a unique molecular pathology profile, which significantly differs from other cutaneous melanomas and is closer to that of uveal melanomas.

## 1. Introduction and Clinical Significance

Nevus is a compound term, originally meaning birthmark. Nevi represent a broad group of conditions resulting from the improper migration or organization of structures and tissues in the skin during embryonic and fetal development. This broad category encompasses multiple distinct entries, including sebaceous nevus (also known as Jadassohn nevus, which involves the sebaceous gland), eccrine and apocrine nevi (sweat gland nevi), connective tissue nevi (such as elastic and collagenous nevi), and melanocytic nevi. Due to the rarity of other types of nevi, the term “nevus” is usually reserved for melanocytic nevi.

Regardless of the types of tissues and cells they comprise, nevi have a somewhat dubious place in human pathology, while ethiopathogenically they are examples of hamartomas—disorganized normal tissues often with atypical locations. Clinically, especially melanocytic ones, behave like benign tumors [1,2]. Melanocytic nevi are rarely visible after birth and usually start to manifest visibly in early childhood, although they can develop throughout life. This is first due to the time required for pigment accumulation, and secondly, the fact that melanocytic nevi often grow over time, bringing their behavior closer to that of tumors. Secondly, similarly to tumors, there is often an immune response from the macroorganism towards them, and some melanocytic nevi can undergo immune-related regression. Lastly, melanocytic nevi are often precursor lesions of malignant tumors, specifically melanoma, despite a not-insignificant number of melanomas developing de novo.

A specific type of rare melanocytic nevi, which recapitulate all of the characteristics mentioned so far, are blue nevi. Blue nevi are dermally located, unlike the physiological placement of melanocytes in the basal layer of the epidermis. However, they retain the normal structure of physiological melanocytes—dendritic and spindle cells, which have prominent melanin pigment production; however, as there are no keratinocytes to which they pass the pigment, the pigment is either retained in the melanocytes themselves or pigment incontinence with deposition in the dermal collagen is present. Blue nevi can be present from birth, such as the nevus of Ota and the nevus of Ito, which further develop in specific topographical areas, underlining their origin as improper melanocyte migration during embryonic and fetal development. The nevus of Ota is located on the face in areas corresponding to the first two branches of the trigeminal nerve, particularly in the periocular area, which is why it they are also referred to as oculodermal melanocytosis, congenital oculodermal melanocytosis, and naevus fuscoceruleus ophthalmomaxillaris. Ota nevi are usually unilateral, but in a small number of cases can be bilateral and symmetric [3,4]. The nevus of Ito, on the other hand, is located on the side of the neck, supraclavicular and scapular areas, and the shoulder region [4].

A third entry into this group is the so-called Mongolian blue spot, also referred to as congenital dermal melanocytosis, which is most often present from birth and located in a specific topographical region—the sacrum and buttocks [5]. As the entries mentioned so far are present from birth and rarely, if ever, lead to malignant transformation, they often present only a cosmetic defect and often spontaneously regress in the school years. They are broadly recognized as dermal melanocytomas, which represent a group of true hamartomas and not melanocytic tumors [5,6,7]. The entries mentioned so far are often grouped as plaque-like blue nevi [8].

The last two entries in the blue nevi category, on the other hand, are recognized as true melanocytic tumors—blue nevi and melanoma arising in blue nevi. Conventional (true) blue nevi are once again melanocytic lesions located in the dermis, usually deep, and recapitulate the physiological morphology of epidermal melanocytes. These lesions are rarely present at birth, but typically develop in young individuals. Clinically, they present as a symmetrical, sharply demarcated lesion, which can develop in any anatomical area, including extracutaneous ones, and have a characteristic blue hue to them, hence the name. Interterminologies used to describe blue nevi are common blue naevus, dermal dendritic melanocytic naevus, naevus of Jadassohn, Tièche naevus, Jadassohn–Tièche blue naevus, cellular blue naevus, and dendritic blue naevus, with the last two depicting the two recognized orophological forms and the previous one serving as either a morphological description or as a historical homage.

Dendritic blue nevi, as their name suggests, comprise dendritic hyperpigmented melanocytes, located deep in the dermis and admixed within a densely fibrotic stroma. Oftentimes, the pigment within the cells is so intensely aggregated that the pigment outlines the cell and completely obscures the nucleus. Cellular blue nevi, on the other hand, do not have such a pronounced dendritic morphology, presenting predominantly with fusiform cell morphology. Pigment production is less pronounced; hence, cell outlines and nuclei are unobscured, and the pigment itself is aggregated in melanophages (melanin pigment macrophages) with a denser cellular population and less pronounced fibrotic stroma.

Last but not least, melanoma arising in blue nevus, also referred to as malignant blue nevus, blue naevus-like melanoma, melanoma ex-blue naevus, and melanoma mimicking cellular blue naevus, is a subtype of melanoma, which is characterized by its deep dermal location, predominantly in areas associated with blue nevi, having a similar color, but a fast-growing lesion comprising malignant spindle cell melanocytes and having a remnant of the preexisting blue nevus. As such, this diagnostic category is reserved for lesions where a conventional preexisting blue nevus can be identified morphologically, or in cases where the tumor developed in the place of a previously excised blue nevus. The diagnosis is often challenging as the melanoma can completely engulf the preexisting blue nevus. In such instances, as well as in other melanomas with marked spindle cell morphology, the term spindle cell melanoma should be applied. Furthermore, melanoma arising in a blue nevus should be differentiated from dermal melanoma metastasis and recurrence, as well as other spindle cell mimickers, such as plexiform melanocytoma, pigmented epithelioid melanocytoma, and clear cell sarcoma, among many others.

## 2. Case Presentation

### 2.1. Cases 1 and 2—Conventional Dendritic Blue Nevus and Sclerotic Dendritic Blue Nevus

A previously healthy 43-year-old female presented to outpatient dermatology consultation due to two pigmented lesions on the dorsal side of the left wrist. The patient could not specify the exact time for which the lesions had been present but stated that they had been present for at least 10 calendar years. The first lesion was topographically located above the first metacarpal bone, was dome-shaped with a blue hue, and measured 6 × 4 mm. The lesion was sharply demarcated with smooth borders. The second lesion was topographically located above the second metacarpal bone, had nearly identical characteristics, and measured 15 mm in diameter. Due to the size of the first depicted lesion and the desire of the patient for its removal, surgical excision was performed. The specimen sent for pathology was represented by a leaf-like skin specimen, measuring 10 × 8 mm, with a centrally placed pigmented lesion of a black-blue color, sharp edges, and a gross distance from the resection margins of 1 mm. On cross-section, the lesion was dermally located and did not involve the deep reepithelial margin. Histology sections revealed non-involved resection margins and a deeply located dendritic melanocyte lesion in the dermis, with significantly colagened stroma and extensive intracellular pigment deposition to the point of nucleus masking (Figure 1). Hence, the diagnosis of dermal blue nevus was established.

The postintervention period for the patient was non-eventful, and the patient was advised to monitor the smaller lesion and have regular outpatient dermatology visits if its characteristics and size started to change.

A previously healthy 48-year-old female presented to our patient surgery consultation due to two lesions on the anterior aspect of the left leg. Both lesions had been present for some time, but the patient could not specify for how long. The first lesion was located proximally behind the tibial head, measured 5 mm, and, upon palpation, was found to be deeply located, relatively mobile, and painful. The overlying epidermis was atrophic and bluish-red, with a gradual gradient towards the non-involved epidermis. The lesion was located in the middle part of the tibial projection, was bluish, and measured 8 mm, had a sharp demarcation, and was non-painful on palpation. As the first lesion was causing discomfort, the patient agreed to the excision of both lesions, under the suspicion of fibrous histiocytomas. The specimens sent for pathology included a leaf-like skin resection with a centrally located and relatively symmetrical, elevated pigmented lesion with an abluish blue border measuring 8 mm. The histopathology of the first lesion confirmed a diagnosis of fibrous histiocytoma (dematofibroma); however, the second lesion exhibited morphological characteristics similar to those of the previously described dendritic blue nevus, with a more pronounced sclerotic stroma. Consequently, the descriptive term “sclerotic blue nevus” was applied (Figure 2).

### 2.2. Case 3—Conventional Cellular Blue Nevus

A previously healthy 55-year-old female presented to outpatient surgery consultation due to a pigmented lesion on the lateral side of the shoulder girdle on the right arm. The lesion measured 8 mm, was symmetrical with sharp outlines, and had a brownish-blue color. Despite the surgeon determining that no extensions were needed, the lesion was excised at the patient’s request for cosmetic reasons. The specimen sent for pathology was a leaf-like spin specimen, measuring 20 × 14 mm, with an acutely located, ovoid, sharply demarcated, and minimally elevated pigmented lesion, which was black–blue and measured 8 mm. Histology revealed a lesion located in the papillary dermis, comprising fusiform melanocytes with minimal intracellular pigment deposition. Melanocytes were arranged in nests surrounded by a rim of sclerotic stroma in which pigmented macrophages were also present (Figure 3). Due to the papillary dermal location of the lesion, artificial pseudopapilliform transformation and atrophy of the epidermis were noted.

### 2.3. Case 4—Cellular Blue Nevus as a Secondary Finding in a Patient with In Situ Clear Cell Squamous Cell Carcinoma and Superficial Spreading Melanoma

A 74-year-old male patient with a previous medical history of moderate hypertension for the past twenty years, under subadequate medication control, with new-onset congestive heart failure for the past five years, type II diabetes under adeqaute oral control for the past fifteen years, and pacemaker implantation the previous year, presented to the surgery clinic following an outpatient dermatology consultation. The patient had multiple facial cutaneous lesions, which were determined to be high-risk and hence suggested for excision. The first lesion was located on the left cheek, measuring 1.5 cm in diameter, and was slightly elevated with smooth borders and a brownish-blue hue. The second lesion was located on the right antihelix and measured 5 mm at its largest, with features of rough and slightly asymmetrical epidermal thickening. The third lesion was located on the posterior section of the scalp, was asymmetrical, slightly elevated above the neighboring epidermis, and measured 1.3 × 0.8 cm. All lesions were excised, with sectioning of the first specimen revealing a demal lesion with a thickness of 1 mm, the second lesion was epidermal in location, and the third lesion had a thickness of 3 mm, and a small neighboring pigmented lesion located dermally with a size of 2 mm. All lesions were grossly excised into healthy tissue.

The lesion on the cheek, histologically, was represented by an atrophic epidermis and a melanocyte lesion with well-defined borders involving the epidermis and extending to the reticular dermis, composed of areas of marked pigmentation and cellularity, with oval to spindle-shaped cells with small monomorphic nuclei, speckled chromatin, and more abundant light or finely pigmented cytoplasm. The tumor cells are arranged in short fascicles and nests, surrounded by pigmented macrophages and circumscriptive sclerotic stroma. Hence, the lesion was diagnosed as a cellular blue nevus (Figure 4). The second lesion histologically was represented by an acanthotic lesion comprising large squamoid cells with light cytoplasm and easily identifiable mitoses, forming intraepidermal nests with microabscesses, single concentric keratin deposits, and focal ulceration. Hence, the lesion was diagnosed as a squamous cell carcinoma in situ (Figure 5). No basement membrane penetration was noted with a dense lichenoid infiltrate located under the lesion. The third lesion was histologically represented by a pigmented lesion with an asymmetric profile, comprising a dermally located neoplastic proliferation of epithelioid-type cells that progressed beyond the border of the dermal component, with nesting structures at the dermoepidermal junction and pagetoid spread. The tumor cells exhibited marked pleomorphism and eosinophilic cytoplasm, as well as vesicular nuclei and prominent nucleoli. An abundance of pigmented macrophages accompanied the sparse intervening stroma, and the pigmentation varied from apigmented to coarsely pigmented. No marked lymphocytic infiltrate, necrosis, or erosion of the surface epithelium was noted. However, the lesion extended to the reticular dermis (Clark IV) and had an overall thickness of 3 mm (Breslow stage III), without epidermal ulceration, the presence of vascular emboli, or atypical mitoses less than 5 per square millimeter (Figure 6). The neighboring pigmented lesion was represented by an epidermally based nevomelanocytic lesion comprising poorly pigmented cells and nests in the dermis, featuring nonpigmented nevomelanocytes with focal low-grade cytological atypia. Hence, the lesion was diagnosed as pT3a melanoma. The patient was referred to the oncology committee for treatment and is currently undergoing treatment and monitoring with stable disease and no progression of either the melanocytic or squamous cell neoplasia.

### 2.4. Case 5—Melanoma Arising in Blue Nevus

A previously healthy 28-year-old male presented to the outpatient surgery consultation after being referred from the dermatology department. The chief complaint is of a rapidly growing pigmented lesion on the middle third, anterolateral aspect of the right leg. The lesion had been present since childhood; however, it recently started growing rapidly. Upon examination, the lesion measured 20 mm × 25 mm and exhibited heterogeneous pigmentation, with intermixing of blue and black–brown areas, as well as a firm, gelatinous rim. The lesion was grossly excised in leathery tissue. The specimen sent for pathology revealed a leaf-like skin lesion measuring 6 × 3 cm, with a centrally located, heterogeneously pigmented lesion measuring 3 cm at its largest. Upon sectioning, the lesion measured 15 mm in thickness.

Histopathology revealed resection margins that were not involved by a tumor process. The lesion presented predominantly with atypical spindle cells of intermediate to low pigmentation, accompanied by minimal stroma, sparse lymphocytic infiltration, and high mitotic activity, including atypical forms, and grew in a nest-like pattern (Figure 7). On multiple sections, both the deep and lateral aspects of the lesion on one of its sides revealed remnants of conventional dendritic blue nevus (Figure 7). The lesson was positive for immunohistochemistry using melanocytic markers, including Melan-A, HMB-45, and S100, and the patient was subsequently diagnosed with melanoma arising in a blue nevus (Figure 8). The patient was referred to the oncology committee for treatment and is currently undergoing second-line treatment due to disease progression.

## 3. Discussion

Blue nevi are by no means rare conditions. They are, however, significantly less common than conventional epidermal, compound, or dermal nevi [9]. Their clinical presentation is well-rounded and is often characterized by well-circumscribed lesions with a bluish hue, which remain stable and may enlarge over time, accompanied by hyperpigmentation. Together, the head and neck regions, as well as the dorsal aspects of the limbs, are often sufficient clinical signs to establish the diagnosis. These lesions are often subject to clinical monitoring only, and not to surgical excision. As seen in the case series presented, patients often remove these lesions for cosmetic purposes, as well as due to multiple excisions of lesions in the same topographical region.

The histological diagnosis of blue nevi can sometimes be challenging. While conventional and sclerotic dendritic blue nevi often pose no morphological challenges due to their exceptional morphology—dendritic melanocytes akin to those physiologically seen in the epidermis and hair follicles, located deeply within the dermis and often in a subcutaneous location as well, and a variably sclerotic stroma. Pigment deposition in these lesions is also excessive, to the point of completely masking cellular and nuclear morphology, with no visible part of the cell structure remaining identifiable due to the accumulation of pigment. Cellular blue nevi, on the other hand, can be more challenging to diagnose. Pigment deposition in this variety can be limited in some cases to the point where these lesions appear to be non-pigmented.

Furthermore, the sclerotic stroma is not intertwined with the melanocytes but surrounds melanocyte nesticles. Together with the fusiform cellular morphology, these can often pose a challenge due to the abundance of spindle cell neoplasms with cutaneous location. Immunohistochemistry with melanocytic markers such as Sox10 and S100 protein is often helpful; however, even in cases with pigment deposition in cellular blue nevi, another differential diagnosis must be made, which includes spindle cell melanoma and dermal metastasis of spindle cell melanoma. In such cases, and especially in cases similar to case 5, wherein the patient has a history of the present diagnosis of melanoma, only extensive morphological evaluation can rule out melanoma metastasis. Key factors in these are cellular morphology, which can also be rare and present with displaced features in blue nevi, and mitotic count, which is not always reliable, as both nevi can have evident mitoses, and melanoma can sometimes lack them. The presence of a variably sclerotic stroma, which is often poor in lymphocytes, is another finding that can be minimal in blue nevi and prevalent in melanoma, and vice versa. To date, the most reliable marker, albeit with some limitations, is BRAF V600E immunohistochemistry, as all blue nevi lack this protein variant. At the same time, at least a not-insignificant percentage of melanomas are positive.

On the other hand, melanoma arising in blue nevi is exceptionally rare and poses the same differential diagnosis challenges. First, they must be differentiated from conventional cellular and pigment-poor cellular blue nevi, which is mainly carried out morphologically, although clinical history, as seen in Case 5, is helpful. In cases of preexisting dendritic blue nevus, however, the differential diagnosis is relatively straightforward. Secondly, they must again be differentiated from metastatic melanoma, spindle cell melanoma, and other categories of spindle cell cutaneous neoplasms. Some gray areas are also present in the application of terminology regarding melanoma arising in blue nevi. While most classifications and authors consider this nosological unit as a remnant of previously present blue nevi in the location of these malignancies, some authors also use the terminology of blue nevus-like melanoma. Blue nevus-like melanoma is a term not recommended by the WHO, nor is the term’ malignant blue nevus’. These entities are descriptive in their terminology and most likely refer to epithelioid and/or spindle cell melanomas. As with blue nevi, melanoma arising in blue nevi is also positive for melanocytic markers and lacks expression of BRAF V600E protein on immunohistochemistry, which, as already stated, is not the most reliable marker, as many melanomas, as well as other spindle cell cutaneous neoplasms, also lack it. Together with melanocytic markers positive by immunohistochemistry and the absence of BRAF V600E positivity, one additional marker on immunohistochemistry can be of aid, and that is the lack of BAP1 [10,11]. Positivity for BAP1 is observed in an overwhelming majority of metastatic melanoma cases, with some studies reporting a figure as high as 95%. On the other hand, and admittedly small and limited in their size, reports support the notion that BAP1 is not expressed in melanoma arising in blue nevus, and state that it is also of aid in differentiating it from non-malignant blue nevi, as BAP1 expression is preserved in them [12,13]. In recent years, PRAME (preferentially expressed antigen in melanoma) has gained popularity. While many melanocytic neoplasms show at least focal positivity, most melanomas show intense diffuse staining with a significant number of differentials of melanoma arising in blue nevus being either negative, or even if positive, not to the extent of most melanomas [14,15,16,17,18,19,20,21].

Another useful marker for differential diagnosis is beta-catenin, as blue nevi and melanoma arising in blue nevus do not produce a positive reaction, while other melanocytic mimickers are generally positive [22]. A summary of useful immunohistochemical markers is provided in Table 1, and of helpful differentiation features in Table 2.

In exceedingly difficult cases, molecular and genomic analysis can also be beneficial, as melanoma arising in blue nevi do not typically exhibit BRAF mutations, but rather GNAQ, GNA11, PLCB4, or CYSLTR2 mutations [23,24]. The reproducibility of this data, however, has yet to be established and will likely be challenging to demonstrate in a large series of cases. This is primarily due to the rarity of melanoma arising in blue nevi and the terminology used by some authors, where the aforementioned epithelioid and spindle cell melanomas are defined as melanoma mimicking cellular blue nevi or blue nevi-like melanomas. Generally, future studies should only include lesions associated with a histologically verified blue nevus in the same area where melanoma development is observed. Lesions with remnant blue nevi should be designated as melanoma arising in blue nevus and related terminologies. As such, a broad and strictly defined cohort can be selected based on these easily applicable morphological criteria, thereby aiding in the identification of immunohistochemical and molecular markers for the diagnosis and differentiation of rare neoplasms.

All of these molecular and genetic features bring melanoma arising in blue nevus closer to uveal melanoma, which also presents with similar morphological features, rather than other cutaneous melanomas [25]. Prognosis of melanoma arising in blue nevus is usually poor, with reported survival being worse when compared to other cutaneous melanomas, once again underlining the unique nature of this rare, pigmented malignancy [26].

### Future Directions

As already discussed, blue nevi and melanoma arising from blue nevus vary significantly in their origin, morphology, and molecular profile when compared to other melanocytic lesions and melanoma. While in the past several years encouraging data has been produced regarding PRAME and its role in the diagnosis of other types of melanoma, similar data is lacking regarding melanoma arising in blue nevus. Future studies would be encouraged to gather such cases and evaluate PRAME expression with the goal of establishing its positivity and possibly producing an immunohistochemical panel for the diagnosis of melanoma arising in blue nevus and its differentiation from other melanocytic lesions and conventional melanoma types.

## 4. Conclusions

Blue nevi with their morphological subtypes—conventional and sclerotic dendritic blue nevi, as well as conventional and non-pigmented cellular blue nevi—are rare compared to other types of melanocytic nevi, yet they remain relatively common throughout the population. They have a somewhat classical clinical appearance and morphology. Melanoma arising in a blue nevus, however, is an exceptionally rare malignancy, to the point that the terminology used in their diagnosis, immunohistochemical markers, and molecular markers remains widely unclear and misunderstood at this time.

## Figures and Tables

**Figure 1 reports-08-00131-f001:**
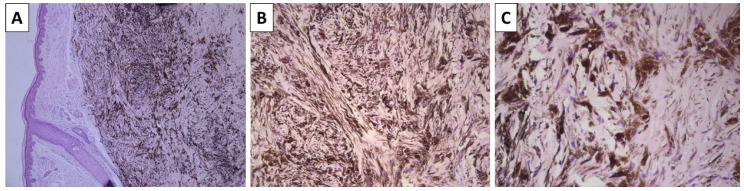
Histopathology of the lesion. (**A**) dermally based pigmented lesion, original magnification 40×; (**B**) spindle-cell and dendritic melanocytes with intense pigment deposition, original magnification 100×; (**C**) dendritic melanocytes are admixed with sclerotic stroma, original magnification 200×; staining (**A**–**C**) hematoxylin and eosin.

**Figure 2 reports-08-00131-f002:**
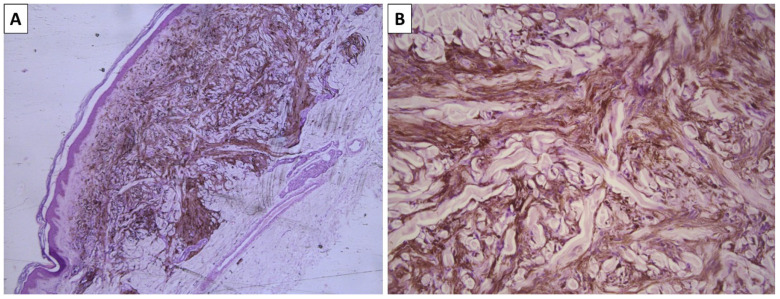
Histopathology of the lesion. (**A**) dermally located pigmented lesion, original magnification 40×; (**B**) dendritic melanocytes with intensive pigment deposition with excessively sclerotic stroma, original magnification 100×; staining (**A**,**B**) hematoxylin and eosin.

**Figure 3 reports-08-00131-f003:**
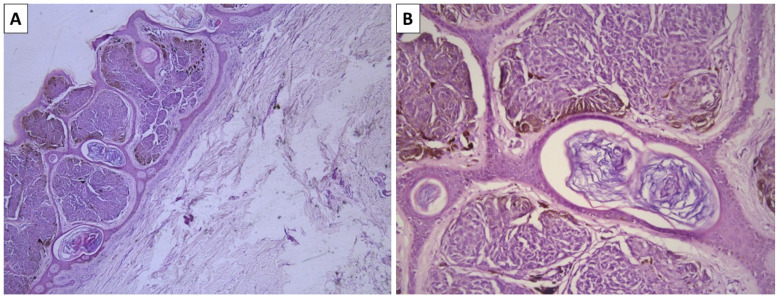
Histopathology of the lesion. (**A**) superficially located dermal melanocytic lesion, original magnification 40×; (**B**) spindle cells melanocytes arranged in nest surrounded by a thin rim of sclerotic stroma, original magnification 100×; staining (**A**,**B**) hematoxylin and eosin.

**Figure 4 reports-08-00131-f004:**
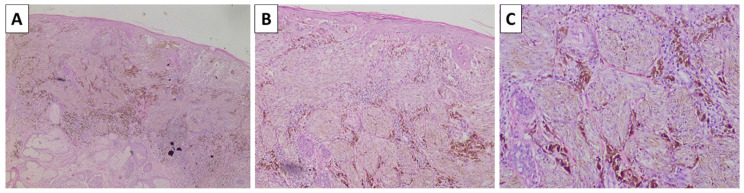
Histopathology of the cheek lesion. (**A**) pigmented lesion in the superficial dermis, original magnification 40×; (**B**) higher magnification showing arrangement of the lesion in nests, original magnification 100×; (**C**) spindle cell melanocytes with subtle pigment deposition and a rim of sclerotic stroma surrounding the nests, original magnification 200×; staining (**A**–**C**) hematoxylin and eosin.

**Figure 5 reports-08-00131-f005:**
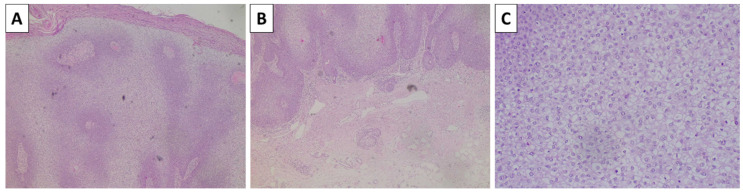
Histopathology of the auricular lesion. (**A**,**B**) acanthotic epidermis with superficial ulceration and no penetration of the basal lamina, original magnification 4×; (**C**) acanthotic proliferation by atypical keratinocyte with clear cell cytoplasmic change, original magnification 100×; staining (**A**–**C**) hematoxylin and eosin.

**Figure 6 reports-08-00131-f006:**
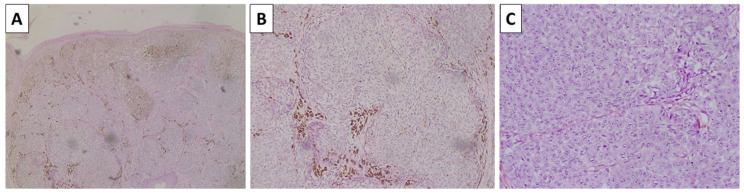
Histopathology of the scalp lesion. (**A**) pigmented lesion with superficial dermal involvement, original magnification 40×; (**B**) large nest of neoplastic melanocytes with subtle pigmentation, original magnification 100×; (**C**) atypical melanocytes with increased mitotic activity, original magnification 200×; stains (**A**–**C**) hematoxylin and eosin.

**Figure 7 reports-08-00131-f007:**
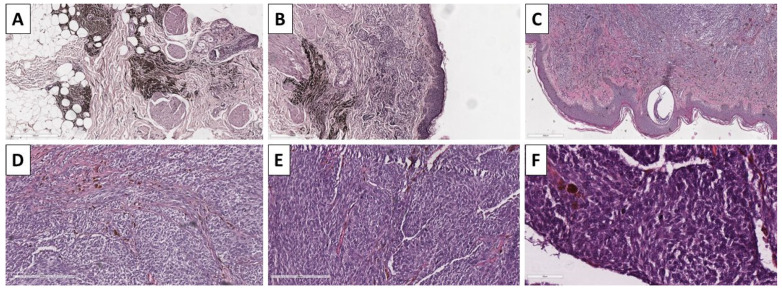
Histopathology of the lesion. (**A**) remnant dendritic blue nevus on the deep resection margin, original magnification 40×; (**B**) remnant dendritic blue nevus and adjacent melanoma nests underneath the epidermis, original magnification 40×; (**C**) subepidermal lesion with subtle pigmentation, original magnification 40×; (**D**,**E**) spindle cell melanocytes with subtle pigmentation and nesticles growth pattern, original magnification 100×; (**F**) increased mitotic activity amongst the nesticles, original magnification 200×; stains (**A**–**F**) hematoxylin and eosin.

**Figure 8 reports-08-00131-f008:**
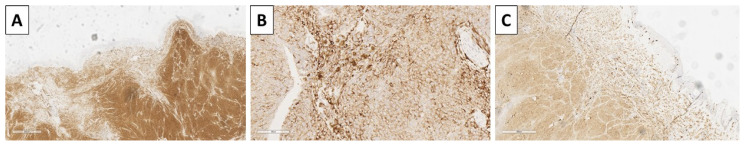
Immunohistochemistry of the lesion. (**A**) Melan-A; (**B**) HMB-45; (**C**) S100; original magnifications (**A**–**C**) 40×.

**Table 1 reports-08-00131-t001:** Immunohistochemical markers helpful in the diagnosis of blue melanocytic lesions; * Small subset of cases can be negative.

Antibody	Expression in Nevi	Expression in Blue Nevi	Expression in Melanoma	Expression in Melanoma Arising in Blue Nevus	Expression in Other Tumors Mimicking Melanocytic Neoplasia
SOX10	+	+	+ *	+ *	+
S100	+	+	+	+	+
Melan-A	+/−	+/−	+/−	+/−	+
HBM-45	+/−	+/−	+/−	+/−	+
BRAF V600E	+/−	+/−	+/−	-	+
BAP1	+	+	+/−	+/−	+
PRAME	+/−	No data	+/−	No data	+

**Table 2 reports-08-00131-t002:** Helpful differentiating features.

Differentiatial Diagnosis	Cellular Blue Nevus	Other Melanoma Types	Melanoma Recurrence or Cutaneous Metastasis	Clear Cell Sarcoma	Other Rare Melanocytic Lesions
Melanoma arising in blue nevus	- Morphology - Mitotic count, especially atypical ones - PRAME staining pattern (likely)	- Melanoma arising in blue nevus is a deeply located dermal tumor - Morphology - Junctional melanocytic component - BRAF V600E - BAP1 staining pattern	- Clinical history - Adjacent scar tissue - Can be challenging if previous histological type is not specified, e.g., recurrence of previous melanoma arising in blue nevus - BRAF V600E - BAP1 staining pattern	- Pigment - Morphology - PRAME rarely diffused positively	- Morphology - β-catenin - PRAME rarely diffused positively

## Data Availability

Data regarding the study is freely available upon reasonable request from the authors.

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
