# Peer review of "Blue Nevi and Melanoma Arising in Blue Nevus: A Comparative Histopathological Case Series"

_reports, 2025, doi:10.3390/reports8030131_

Round 1

Reviewer 1 Report

Comments and Suggestions for Authors

The authors have provided a very detailed pathological comparison between blue nevi and melanoma arising in blue nevi. They highlight key findings and describe each case with great care. My comment is as follows,

1) To make the differential diagnosis more understandable, would it be possible to summarize the key diagnostic features in a comparative table?

2) Recently, the utility of PRAME expression in the differential diagnosis of malignant melanoma has been increasingly reported. Has PRAME immunostaining been evaluated in Cases 1–5? Given its emerging role in distinguishing malignant melanoma, I would encourage the authors to consider including this information in the manuscript.

3) In addition, clinical photographs and dermoscopic images are essential for the differential diagnosis of pigmented lesions. If available, could the authors include such images for Cases 1–5?

Author Response

Point-by-point response to reviewer and editorial comments

Dear editors, staff and reviewers of Report, thank you for considering our manuscript entitled “Blue Nevi And Melanoma Arising in Blue Nevus: A Comparative Histopathological Case Series”; the following is our point-by-point response to the comments made by the respected editor and reviewers:

Comments of reviewer 1

The authors have provided a very detailed pathological comparison between blue nevi and melanoma arising in blue nevi. They highlight key findings and describe each case with great care. My comment is as follows,

  • To make the differential diagnosis more understandable, would it be possible to summarize the key diagnostic features in a comparative table?

  • Dear reviewer, thank you for this valuable suggestion, a corresponding table will be made in the discussion section of the revised manuscript

  • Recently, the utility of PRAME expression in the differential diagnosis of malignant melanoma has been increasingly reported. Has PRAME immunostaining been evaluated in Cases 1–5? Given its emerging role in distinguishing malignant melanoma, I would encourage the authors to consider including this information in the manuscript.

  • Dear reviewer, thank you for this valuable comment. Sadly we have not performed PRAME IHC testing in the cases as most of them are several years old and even in the newer cases we sadly have no access or experience with this antibody. A section discussion PRAME expression in melanoma and its role in diagnosis and treatment will be added in the discussion section of the revised manuscript.

  • In addition, clinical photographs and dermoscopic images are essential for the differential diagnosis of pigmented lesions. If available, could the authors include such images for Cases 1–5?

  • Sadly, we have no access to these. As the respected editor has also requested the edition of these, the following is a copy of our response provided to the respected editor in this query:

Dear editor, thank you for this note. We realize the addition of such data would greatly improve the impact of the manuscript; we sadly, however, have no such photographs as the patients that presented from outpatient dermatology were either advised for excision or the excision was performed as per the patients desires – as such we have no gross photos of the lesion. Surgical staff at our institutions also rarely, if ever, take gross images of cutaneous lesions as skin excisions are organized in so-called “skin days”, where multiple patients, often more than 10 per day are scheduled for excision of cutaneous lesions under local anesthesia only, with the goal of all residents being able to observe a variety of lesions and localization being excised in the same day, when all residents are present. As such due to the “conveyer” line of patients, once again preoperative images are rarely taken. We hope the lack of gross images prior to excision will not hinder the chances of our manuscript being published. Furthermore, as we realize the manuscript is lacking in this area we have specifically focused our attention on histological and molecular features, which is also reflected in the title on the manuscript – “A Comparative Histopathological Case Series”

Reviewer 2 Report

Comments and Suggestions for Authors

Congratulations to the authors! This manuscript presents a case series of blue nevi and melanoma arising in blue nevi, supported by detailed morphological descriptions and histological imaging. This highlights the diagnostic challenges of distinguishing benign blue nevi from melanoma arising in these lesions, and describes clinical and immunohistochemical features.
I would suggest inserting a table of immunohistochemistry markers used for the confirmation of the diagnosis. 
Careful with the typos, some areas require attention:

line 55- romal structure?
line 64- for which reason is also referred to as...would sound better to say which is why it is also referred to as..
line 90- oftentimes the pigment within the cells is so intensely aggregated they the pigment outlines the cell completely -please revise this phrase

Author Response

Point-by-point response to reviewer and editorial comments

Dear editors, staff and reviewers of Report, thank you for considering our manuscript entitled “Blue Nevi And Melanoma Arising in Blue Nevus: A Comparative Histopathological Case Series”; the following is our point-by-point response to the comments made by the respected editor and reviewers:

Comments of reviewer 2

Congratulations to the authors! This manuscript presents a case series of blue nevi and melanoma arising in blue nevi, supported by detailed morphological descriptions and histological imaging. This highlights the diagnostic challenges of distinguishing benign blue nevi from melanoma arising in these lesions, and describes clinical and immunohistochemical features.

  • Dear reviewer, thank you for this kind comment.

I would suggest inserting a table of immunohistochemistry markers used for the confirmation of the diagnosis.

  • Dear reviewer, thank you for this suggestion, a table with relevant antibodies will be included in the discussion section of the revised manuscript

Careful with the typos, some areas require attention:

line 55- romal structure?

line 64- for which reason is also referred to as...would sound better to say which is why it is also referred to as..

line 90- oftentimes the pigment within the cells is so intensely aggregated they the pigment outlines the cell completely -please revise this phrase

  • Dear reviewer, thank you for noticing these typographical errors on our behalf, they will be corrected in the revised version of the manuscript